# A Delphi Study to Determine International and National Equestrian Expert Opinions on Domains and Sub-Domains Essential to Managing Sporthorse Health and Welfare in the Olympic Disciplines

**DOI:** 10.3390/ani13213404

**Published:** 2023-11-02

**Authors:** Jane M. Williams, Lise C. Berg, Hilary M. Clayton, Katharina Kirsch, David Marlin, Hayley Randle, Lars Roepstroff, Marianne Sloet van Oldruitenborgh-Oosterbaan, Michael A. Weishaupt, Carolien Munsters

**Affiliations:** 1Equine Department, Hartpury University, Gloucester GL19 3BE, UK; 2Department of Veterinary Clinical Science, University of Copenhagen, Hoejbakkegaards Alle 5, 2630 Taastrup, Denmark; lcb@sund.ku.dk; 3Department of Large Animal Clinical Sciences, Michigan State University, East Lansing, MI 48824, USA; claytonh@msu.edu; 4Department Sensors and Modeling, Leibniz Institute for Agricultural Engineering and Bioeconomy (ATB), Max-Eyth Allee 100, 14469 Potsdam, Germany; kkirsch@atb-potsdam.de; 5AnimalWeb Ltd., Tennyson House, Cambridge CB4 0WZ, UK; dm@davidmarlin.co.uk; 6School of Agricultural, Environmental and Veterinary Sciences, Charles Sturt University, Wagga Wagga, NSW 2678, Australia; hrandle@csu.edu.au; 7Department of Anatomy, Physiology and Biochemistry, Faculty of Veterinary Medicine and Animal Sciences, Swedish University of Agricultural Sciences, SE-75007 Uppsala, Sweden; lars.roepstorff@slu.se; 8Department of Clinical Sciences, Faculty of Veterinary Medicine, Utrecht University, 3584 CM Utrecht, The Netherlands; m.sloet@uu.nl; 9Equine Department, Vetsuisse Faculty, University of Zurich, 8057 Zurich, Switzerland; weishaupt@vetclinics.uzh.ch; 10Equine Integration B.V., Groenstraat 2c, 5528 NS Hoogeloon, The Netherlands; carolien@munsters.nl

**Keywords:** welfare, equine management, equine training, equestrian, social license to operate, horse sports, dressage, showjumping, eventing, horse–rider relationship

## Abstract

**Simple Summary:**

Horse sports are popular worldwide, providing spectator enjoyment, benefiting human health, and contributing substantially to national economies. Training and management practices used to care for sporthorses are generally based on tradition rather than science; this combined with the high-risk nature of equestrian activities led to the public questioning if sporthorse health and welfare are being compromised. To understand better how sporthorses are being managed, experts, actively involved with national and international horse sports (dressage, showjumping, and eventing) were consulted across four rounds of a Delphi study. This approach allowed participants to interact to reach a point where everyone agreed on core areas (or domains) that they felt were essential to sporthorse management. Five areas were rated as essential: training management, competition management, young horse management, health status and veterinary management, and the horse–human relationship. Stable and environmental management, and welfare assessment were rated as important but not essential, as most experts felt that these areas were already managed well. Experts called for increased education and research to support riders, trainers, and federations. A welfare charter and evidence-based guidelines to inform management practices were advised to ensure sporthorses have a good life and to safeguard the future of equestrian sports.

**Abstract:**

The public is increasingly questioning equestrianism’s social license to operate. While the focus historically centered on horseracing, increased scrutiny is now being placed on how dressage, showjumping, and eventing are addressing equine management and welfare concerns. Nominated equestrian federation and equestrian organization experts (*n* = 104) directly involved in international and/or national-level horse sports took part in a four-stage, iterative Delphi to obtain consensus on what factors should be considered essential to manage sporthorse health and welfare. Five core domains were agreed as essential: training management, competition management, young horse management, health status and veterinary management, and the horse–human relationship. Two further domains: stable and environmental management, and welfare assessment were rated as important but not essential, as most respondents felt that these areas were already managed well. Participants felt increased education and guidance combined with further policy development and regulation are needed to support stakeholders to optimize sporthorse management. An appetite to engage with research to generate evidence that promotes sporthorse welfare was evident. The development of a sporthorse welfare charter and evidence-based guidelines to inform the management and monitoring of sporthorses’ health and welfare are recommended to provide horses with a good life and to safeguard the future of equestrian sports.

## 1. Introduction

Horse sports are popular worldwide and provide spectator enjoyment, benefits to human mental and physical health, and contribute substantially to national economies [1,2]. However, the high-risk nature of equestrian activities and the potential of injury or fatality in human and equine participants, alongside increasing public scrutiny of the potential impact of equine management and training practices on equine welfare, resulted in equestrianism’s social license to operate (SLO) being questioned [1,3,4]. High-profile horse fatalities, examples of poor horse welfare at leading events, and television documentaries highlighting negative practices in the industry (e.g., The Final Race, Australia; The Dark Side of Racing, UK) reported in the mainstream press and openly discussed on social media channels fuel this debate [4]. Until recently, societal concern was focused on the welfare of racehorses due to the visibility and profile of the sport in the public domain. Concern now shifted to debate the welfare of horses across all equestrian disciplines [5,6]; in particular, the management of sporthorses, especially those competing at international levels, such as performing in the Olympic and World Championship in dressage, showjumping, and eventing. Examples include the Arjen Lubach ‘Avondshow’, Netherlands, Dier&Recht petition requesting a ban of coercive training methods (e.g., bits and bridles) to the Dutch Staten-Generaal [7]).

How horse welfare is valued is primarily determined by the people who manage horses and the decisions that they make [4,8]. Rapidly changing societal values called into question many traditional equestrian practices, such as restricted turnout, and equestrianism’s SLO [3,8]. Public opinion may be clouded by anthropomorphism or ignorance of the needs of different species, leading to emotional rather than objective decision-making. Unfortunately, limited objective evidence exists to justify many common practices used in horse sports that are currently perceived as detrimental by the public; for example, whip use [8,9,10]. Nevertheless, the Sporthorse Welfare Foundation agrees with Campbell [1] that the use of horses in sport is ethically justifiable, if an ethical framework can be applied to safeguard horse welfare by supporting stakeholders to make evidence-informed decisions on what should or should not be carried out in specific situations [8,11]. These decisions can then be articulated to the public to demonstrate how those involved with equestrian sport are managing horse welfare effectively and meeting SLO expectations [12,13].

An SLO is earned and should provide the framework for an industry, sector, or sport to define the boundaries within which it operates. It usually builds on trust (procedural), operational transparency, and communication to confirm operational legitimacy, which then provides the credibility necessary to enable society to accept that a SLO is embedded within it [3,12,13]. Social license should be viewed as a dynamic concept which, by its nature, will evolve and develop as the sector it applies to progresses, thus reinforcing its credibility and generating trust from society that the framework which governs it is worthy of continued approval [13,14]. Within equestrianism, this translates to the scrutiny of human interactions with horses, alongside analysis of the management, use, and care applied to provide horses with a good quality of life. Within the global equestrian industry, there is a need to be transparent and prove to the public how the physical and psychological needs of the sporthorse are managed to optimize health and welfare. However, despite increased debate around equestrianism’s SLO, there is still a lack of empirical evidence available to show how sporthorse health and welfare are managed on a daily basis. There is also a lack of insight into equestrians’ knowledge and perceptions of equine health and welfare, and the differences and inter-relationship between these [1,2].

In an area where public emotions run high, such as the evaluation of animal welfare, objective evidence regarding current opinion is required to provide a foundation that informs future progress. The management of horses ultimately depends on the people who care for them. Recent attention to SLO highlighted areas of deficit resulting in compromised or suboptimal welfare (such as prolonged stabling with limited opportunities for social interaction between horses) [2]. It is well known that horse management is embedded in tradition and that there is substantial resistance to change across the sector [8,15]. However, if public concerns are to be addressed, there is a fundamental need to understand human behavior in relation to horse management practices and how this informs decision-making before seeking to make changes that will improve horse welfare [4,16]. Models of human behavior suggest that to change behavior, an appropriate first step is to seek to understand existing attitudes to the behavior/s in question. The next step is to ascertain who is and is not performing the behaviors, when and how often, and to what extent political, social, and environmental influencers underpin existing decision-making [16].

### 1.1. The Delphi Method

The Delphi technique [17] is a well-established robust scientific approach to answering a research or practice-based question through the identification of a consensus view across subject experts [18]. Using this method, invited expert panel members are able to review and revise their responses across each stage of the survey. The controlled feedback process provides anonymity to the respondents and suppresses the influence of domineering individuals and opinions, which may be a factor in group-based discussions [18,19,20]. Participation of between 20 and 30 experts is considered to be representative of a field [20,21].

Delphi studies outline the opinions of the people or ‘experts’ selected to take part in them. For this reason, they are common in health research where knowledge is incomplete or where different opinions may exist to reach a consensus on which clinical guidelines or performance indicators to use. They were used to determine which stages of a clinical protocol are essential or critical for a successful outcome. For example, they were successfully used to identify diagnostic indicators and treatment regimens for cancer [22,23], dementia [24,25], and pain [26,27]. They were also used across animal species to define acceptable and priority measures for animal welfare [28,29,30,31].

The experts are asked their opinion across domains and about core attributes that contribute to the area under consideration [18]. Expert agreement is gathered over a number of consultative stages until agreement between the participants’ responses is reached, with the aim of achieving consensus [18,20]. It is important to remember that a Delphi consultation is an interactive and iterative process, shaped at all stages by the participants, with the final result summarizing the views of the participants, not those of the researchers. The desired outcome is to reach a consensus or agreement across the experts to clarify the essential and non-essential components of the area being evaluated, but there is the potential for no consensus to be attained and for areas to be omitted if they are not raised by the participants [20].

### 1.2. Managing Sporthorse Health and Welfare

Equestrian sports include horses as an essential athletic partner for the human athletes competing in them. Worldwide, there are many equestrian sport disciplines; for example, horseracing, polo, and Western riding, but only a few are performed at the Olympic level (dressage, showjumping, eventing, and para dressage) and/or World Championship level (dressage, showjumping, eventing, endurance, driving, vaulting, para dressage, para driving, and reining). Some of these are governed globally by the Fédération Equestre Internationale (FEI). National-level governance is provided by country-specific equestrian federations who provide a vision and leadership and determine the future direction of horse sports. It is important to note that there are many other sports involving horses, such as polo and showing (showing as a discipline varies across countries; it can include in-hand and ridden classes assessing conformation, ridden expertise and potential use, equitation prowess, breed classes, or Western showmanship), but whilst these areas have associated ‘organizing bodies’, they may not be considered ‘sporthorse’ disciplines in a similar way to those recognized at Olympic and World Championship levels and are viewed more as associations. These national sporthorse equestrian federations and higher-level organizations provide regulatory guidance for how the FEI-governed horse sports are managed in competitive environments from elite to grassroots levels. Although, the majority of national federations recognized the strategic importance of promoting frameworks that support equine welfare and safeguard the future sustainability of horse sports, how this translates to member practice remains relatively unknown.

The aim of this study was therefore to establish existing opinions in these horse sports by asking equestrian federations and organizations to nominate global equestrian stakeholders with different roles in the industry who were actively involved in managing horse sports for the Olympic disciplines: dressage, showjumping, and eventing, at an international and/or national level, to provide their views/opinions on what factors should be considered essential to the management of sporthorse health and welfare.

## 2. Materials and Methods

### 2.1. Determining Domains Essential to Manage Sporthorse Health and Welfare

#### 2.1.1. Recruitment Strategy

A four-round, iterative Delphi was performed to systematically develop consensus among key equestrian expert groups across a 6-month period. To be eligible to participate in the Delphi, inclusion criteria required that countries took part in and placed as a team in at least one equestrian discipline/sport in the last five Olympic Games. Thirty-two equestrian federations were invited to take part in the study. In addition, three overarching equine welfare, sport, and research organizations participated: World Horse Welfare, the International Dressage Trainers Club, and the Sporthorse Research Foundation.

Each national federation or organization was asked to nominate participants based on their expertise on health and welfare of sporthorses performing at Olympic and/or international level [20,21]. Participants represented the interdisciplinary opinion of five groups of equine professionals involved with managing sporthorse health and welfare at a national federation or international equestrian organization level:Equine veterinarians.Allied professionals e.g., farriers, physiotherapists.Olympic/International level coaches/trainers OR professional, international riders.Equine welfare experts OR key federation employees.Equine researchers.

Of the 32 equestrian federations contacted, 24 federations, World Horse Welfare, the International Dressage Trainers Club, and the Sporthorse Research Foundation nominated 117 participants to take part in the study; although it should be noted that not all invited participants input into all stages of the study (refer to Figure 1 for details). Where federations did not respond, relevant experts from across the participant groups, who met the inclusion criteria, were contacted directly to participate in the study. Nominees were contacted by email to confirm their consent and invited to participate in the study. Participation was voluntary and no financial compensation was provided.

The moderator (JMW) communicated with each expert on an individual basis and recorded responses anonymously on a master datasheet [21]. For Stage One, questionnaires were distributed in English; however, from Stage Two onwards, in response to respondent feedback, questionnaires were translated into French, Dutch, German, and Spanish, and individuals could select the language they completed the survey in. Responses not in English were translated to facilitate analysis.

The term *domain* was defined as an overarching area that should be included within a framework or set of guidelines to manage the health and welfare of horses performing in the Olympic disciplines: dressage, showjumping, and eventing. A *sub-domain* was defined as a topic that would contribute to the effective management of a domain. At this stage, the number of survey rounds was not fixed and was to be determined by the degree of consensus within the panel of experts. We did, however, expect that there would be three to five rounds, with the last round providing a final opportunity for the experts to revise their judgments [18].

#### 2.1.2. Delphi Stage One

An email with a link to a questionnaire (Qualtrics XM™, Seattle, WA, USA) was sent out to the experts that responded positively to being included in the Delphi panel. This stage aimed to assist selection of the domains that should be included in the final consensus statement [20,32]. Domains included in the first iteration were developed by the research team based on existing literature in equine health and welfare relevant to sporthorse management [20,32]. During this first stage, the experts were asked to rate whether twelve areas: (1) training management; (2) competition management; (3) biomechanical/locomotion assessment; (4) stable and environmental management; (5) behaviour; (6) young horse management; (7) health status and veterinary management; (8) nutrition; (9) use of allied professionals; (10) horse–human relationship; (11) judges, officials, and rules; and (12) welfare assessment, were essential, not essential but important, or not essential to manage sporthorse health and welfare [21,33,34]. Experts could also suggest additional domains that could be included, and they were asked to identify potential sub-domains that should be included within these domains to manage them effectively in order to inform questionnaire design for Stage Two [33]. 

The concept of consensus across the expert group was defined as a condition of the consistency of opinion found between respondents to represent the interdisciplinary consensus of the experts working across different elements of horse sport that came together in the present study [21,33]. Consensus in this study was defined as agreement by a minimum of 70% of the experts (referred to as the ‘critical value’) [35,36,37]. To further examine the extent of consensus within and between the experts, three statistics were utilized.

1.Content validity ratio

From the responses gained, a content validation calculation was used to agree to include or discard items listed as possible domains [20,38], with content validity ratio (CVR) and critical values used to confirm the level of agreement that exceeded that of chance [34]. Each expert could rate the domains as (1) essential, (2) not essential but important, or (3) not essential. The CVR was calculated as:$$CVR=\frac{ne-\frac{N}{2}}{\left(\frac{N}{2}\right)}$$
where “CVR”: content validity ratio, “*ne*”: number of essential members, and “*N*”: number of panel members [33,38].

Perfect agreement results in a CVR of +1 and perfect disagreement results in a CVR of −1. Agreement of >±0.7 (corresponding to 70% of the experts) was deemed appropriate to confirm expert consensus that a domain/sub-domain was essential within the management of sporthorse health and welfare.

2.Content validity index

To ascertain the level of agreement across the expert panel, average agreement was also calculated using the content validity index (CVI) *a posteriori* to evaluate the reliability of the consensus obtained through expert assessment [39,40]. The CVI reflects the average agreement rated as a percentage and calculated across all the core domains and across all factors within each domain. To obtain the CVI/average agreement for relevancy and clarity of each item, the number of those judging the item as essential (CVR) was divided by the total number of domains/subdomains [41]:$$CVI=\frac{\sum CVR}{n}$$
where “CVR”: content validity ratio value for individual domains or sub-domains in a defined area, and “*n*”: number of sub-domains within the defined area.

Again, a threshold value of >±0.7 was applied to indicate that CVI values had exceeded the threshold to confirm consensus of expert opinion, i.e., that a domain/sub-domain was essential within the management of sporthorse health and welfare. CVR values that exceeded the calculated CVI (i.e., the mean value) for a domain/sub-domain were rated as achieving average agreement that a domain/sub-domain was essential within the management of sporthorse health and welfare. For CVR values that did not attain the average agreement (CVI) threshold for a domain/sub-domain, this was interpreted as a lack of consensus for the respective domain/sub-domain [41]. For example, if the CVI value for a domain was determined to be 0.56, then factors whose values were between 0.57 and 0.69 would demonstrate above-average agreement, while those <0.55 would be considered to have a lack of consensus.

3.Cronbach’s Alpha

The internal consistency within the panel of experts between each round was assessed using Cronbach’s coefficient alpha [21,42], as follows:$$\mathrm{Cronbach}’s\;\mathrm{coefficient}\;a=\frac{n}{\left(n-1\right)(1-\sum \frac{i\sigma 2i}{\sigma 2t}\;)}$$
where “*n*”: total number of experts, *σ*2: variance of each individual expert response, and *σ*2*t*: variance of the sum responses for each individual expert [21,42].

This was rated in line with previous Delphi research [21,43], with an overall Cronbach’s coefficient α value higher than 0.8 considered as a threshold demonstrating excellent internal consistency; 0.70–0.79 good internal consistency; 0.60–0.69 moderate internal consistency; 0.50–0.59 not sufficient internal consistency; and less than 0.50 low internal consistency.

4.Content analysis

For open questions in each stage of the Delphi, and subsequent seminars, conventional, inductive content analysis of respondent comments and views was undertaken to identify core areas and themes which emerged from these [33,44].

5.Delphi Stage Two

In Stage One, the experts agreed on seven domains and proposed no new domains for inclusion in Stage Two. For the remaining core domains, definitions were generated from expert feedback. Participants were then asked to rate their agreement for the seven core domains and their definitions. In addition, sub-domains, proposed by participants in Stage One, were listed under each core domain, and the experts were asked to rate whether these topics were essential to manage sporthorse health and welfare, using a more specific Likert scale (0: not essential to 9: always essential) to allow for more nuanced opinions [45,46]. Answers scoring 7, 8, or 9 on the scale were deemed to exceed the threshold for an essential rating to facilitate CVR and CVI calculation as outlined in Stage One.

6.Delphi Stage Three

Participants were asked to identify and agree what tools and measures were currently available and should be used to monitor and manage sporthorse health and welfare, using the same Likert scale as Stage Two (0: not essential to 9: always essential). Answers scoring > 7 on the scale were deemed to be the equivalent of an essential rating to facilitate CVR and CVI calculation as outlined in Stage One.

7.Delphi Stage Four

Consultation (DM, CM, JMW, LB, and MW) occurred with the experts through online webinars prior to Stage Four to provide the opportunity for participants to raise any questions or queries across any of the areas and topics included. The list of domains and sub-domains that met the agreement criteria were emailed to the panel of experts who were invited to confirm the final selection [18,20,33]. Participants were also asked to propose potential reasons why some areas and topics were not selected, as well as their opinions on the future of horse sports and how the results of the Delphi should be used.

Across all rounds of the Delphi, respondents were encouraged to comment on the core areas and topics and add any additional areas they felt were important; selection of the final areas and topics was based on both ratings and these comments [20]. Figure 1 summarizes the Delphi process undertaken.

8.Differences by expert role and country

Frequency analysis identified the percentage of experts who rated specific domains and subdomains less than four (i.e., rating the specific area ‘*as not essential*’ for sporthorse health and welfare) across all countries that participated in the Delphi and within each expert group. A series of Kruskal–Wallis analyses evaluated if individual expert experience or country represented influenced expert ratings across domains and subdomains. Where significant results were found, post hoc Mann–Whitney U analyses identified where differences existed between the expert groups assessed. Significance was set at *p* < 0.05.

## 3. Results

While all stages of the study exceeded the levels required to be representative (Figure 2), some countries did not take part; these were mainly countries that participated at the Olympics only once. The 24 countries that participated in the study represent key countries that were present and placed at recent Olympic Games. Figure 3 outlines the distribution of roles across the experts who participated in the Delphi study.

Across the Delphi, consistent ratings were reported within the same individual experts between rounds (i.e., the same person consistently recorded the same opinion in each stage of the Delphi), but a wide range of ratings between individual experts occurred, representing variable opinions across the expert group as a whole. Some individual experts rated domains and sub-domains as essential and consistently high, while others rated them consistently low (Table 1).

### 3.1. Consensus: Core Domains of Sporthorse Health and Welfare Management

After the four rounds, there was 100% agreement across the experts that the domains ‘training management’, ‘competition management’, ‘young horse management’, ‘health status and veterinary management’, and the ‘horse–human relationship’ were ‘*essential*’ for sporthorse health and welfare (Table 1). Within these core domains, young horse management received the highest level of expert agreement (CVR of 0.89) that it was ‘*essential*’ for sporthorse health and welfare. Training, maintaining a healthy horse, and horse–human relationship were also generally rated highly. Interestingly, regardless of the level of agreement for the core domains, experts often rated the sub-domains, presented in Table 2, Table 3, Table 4, Table 5, Table 6, Table 7 and Table 8, which underpinned these areas ‘*as not essential’*.

While definitions were agreed for stable and environment management and welfare assessment, these areas were not rated as essential areas within the management of sporthorse health and welfare. These results were unexpected; therefore, the experts were questioned in Stage Four to try to understand the reason for this. Sixty-one percent of experts rated stable and environment management as ‘*important but not essential*’, as they felt it was already managed well, and 39% felt it was ‘*important but not essential*’ as it was less crucial for sporthorse welfare. Three key themes emerged which may explain why this area was not agreed as essential:Perception can be this area is well managed, but in practice management is variable.[Horses’] Mental wellbeing and behavioral needs can be difficult to accommodate.Practice is mixed: A strong framework for stable and environment management exists in some countries but regulation could help in this area.

A similar pattern occurred for welfare assessment; 71% of experts rated welfare assessment as ‘important but not essential’ as it is already managed well, 25% felt it was ‘important but not essential’ as it was less crucial for sporthorse welfare, and 4% did not feel it was important or essential for sporthorse welfare. Three key themes emerged which may explain why this domain was not agreed as essential:Welfare should be integrated into all aspects of sporthorse management and is not a standalone area.Welfare is already well managed/regulated and therefore perception is this area is not as essential (despite variable knowledge, understanding, and practice).Defining how to assess [horse] welfare is difficult.

### 3.2. Consensus: Sub-Domains That Underpin Core Areas of Sporthorse Health and Welfare Management

Although five domains were rated as ‘essential’, generally, there was a lack of agreement for which sub-domains within these areas were essential for sporthorse health and welfare. Across the results, the consistency of agreement was high overall (>0.92 in essential domains, >0.83 in above average agreement for sub-domains within these domains); however, the average agreement for whether a domain or sub-domain was essential or not for sporthorse health and welfare varied between participants.

Overall, 30% (±17%, range 8.3–42%) of all sub-domains were rated as ‘essential’, 27% (±20%) had above-average agreement, and 43% (±9%) lacked sufficient agreement to be regarded as essential to manage sporthorse health and welfare. Therefore, although there was consensus on what the essential core domains were, expert views on which of the sub-domains underpinning these domains were needed to manage them effectively was mainly unclear.

#### 3.2.1. Training Management

Agreed definition: Management of structured, evidence-informed activities that apply knowledge, skills, tools, and techniques to training activities to achieve targeted performance outcomes. This should include ensuring the horse is physically (has an appropriate level of skills, fitness, strength, and condition) and mentally prepared for exercise and all aspects of competition. The potential benefits include reducing injury risk while retaining high standards of equine welfare.

Training Management was rated highly with 0.78 agreement that this area is ‘*essential*’ for the management of sporthorse health and welfare (Table 1). However, from the 18 sub-domains proposed, consensus on which areas were essential in managing sporthorse health and welfare was agreed for only two sub-domains (11% of all sub-domains). These were monitoring career longevity and training environment. Ten sub-domains (56%) recorded above-average agreement, and six (33%) lacked consensus including recovery, training program, and monitoring of training (Table 2). In Stage Four of the Delphi, the majority (79%) of stakeholders felt that sub-domains that scored lower in training management were ‘*important but not essential*’ as they were already managed well; while 21% rated these areas as ‘important but not essential’ as they were less crucial for sporthorse health and welfare.

#### 3.2.2. Competition Management

Agreed definition: management of horses directly before, during, and after a competition; includes travel, stabling, nutrition, biosecurity, exercise, and wellbeing management to optimise health, performance, and welfare.

Within the core domain of competition management, ‘health monitoring’ was the only sub-domain agreed by participants to be ‘*essential*’ to sporthorse health and welfare, 34% of the sub-domains recorded above average agreement, and 58% lacked consensus (Table 3). In Stage Four of the Delphi, the majority (73%) of stakeholders felt that sub-domains that scored lower in competition management were ‘*important but not essential*’ as they were already managed well; 24% rated these as ‘*important but not essential*’ as they were less crucial for sporthorse health and welfare, and 3% felt they were not important, as they were not as crucial for sporthorse health and welfare.

#### 3.2.3. Young Horse Management

Agreed definition: management of physical and mental wellbeing to optimise the growth, health, and future performance of the young horse from the start of the breeding process to the introduction of training and competition as a sporthorse.

While all experts agreed that core domain ‘young horse management’ was ‘*essential*’ for the management of sporthorse health and welfare, there was little consensus on which sub-domains were essential within this domain. Only 31% of sub-domains were defined as ‘*essential*’, 38% had above-average agreement, and 31% lacked consensus (Table 4). The majority (72%) of stakeholders felt that sub-domains that scored lower in young horse management were rated as ‘*important but essential*’; because they were already managed well, 21% rated these as ‘*important but not essential*’ as they were less crucial for sporthorse health and welfare, 3% felt they were not important as they were already well managed, and 3% felt they were not important as they were not as crucial for sporthorse health and welfare.

#### 3.2.4. Health Status and Veterinary Management

Agreed definition: Provision of veterinary and allied professionals care, health, and training management that prepares the horse for training and competition and reduces the risk of injury and disease. This should include regular monitoring of fitness, respiratory, cardiovascular, and orthopaedic health, body condition and nutritional management, behaviour, and the mental wellbeing of the horse.

Across the core domains agreed, health status and veterinary management recorded a high number of sub-domains, a total of 12 areas. Experts agreed 42% of these were ‘*essential*’, 16% had above-average agreement that they were ‘*essential’*, and 42% lacked agreement (Table 5). The majority (89%) of stakeholders felt that areas that scored lower were ‘*important but not essential*’ as they were already managed well, 23% rated these as *’important but not essential*’ as they were less crucial for sporthorse health and welfare, and 8% felt they were not important as they were not as crucial for sporthorse health and welfare.

#### 3.2.5. Horse-Human Relationship

Agreed definition: Assessment of interactions between horses and humans. There may be a wide range of types of interactions of varying intensity, which should be monitored under various conditions, such as when being handled, trained, and worked at home or at competition.

Approximately one third (36%) of sub-domains reached consensus that they were ‘essential’ to the management of sporthorse health and welfare, 9% had above-average agreement, and 55% lacked agreement (Table 6). Interestingly, assessment of human–horse interactions was rated higher as being essential than the relationships underpinning these interactions between the horse and the rider, groom, and coach/trainer, which did not reach consensus. The majority (68%) of stakeholders felt that lower-scoring areas were ‘important but not essential’ as they were already managed well, 25% rated these as ‘important but not essential’ because they were less crucial for sporthorse health and welfare, 4% felt they were not important as they were already managed well, and 4% felt they were not important as they were not as crucial for sporthorse health and welfare.

#### 3.2.6. Stable and Environment Management

Agreed definition: Management of the housing and external environment of the horse. This includes provision and management of stabling, turnout, social interaction with and between horses, exercise facilities, nutrition, water, veterinary care, and management schedules to optimise horse health and welfare, as well as horse management in response to changes in the external environment or climate.

Stable and environment management was not agreed as a core domain for the management of sporthorse health and welfare; 25% of sub-domains were agreed as ‘essential’ to manage sporthorse health and welfare, 25% had above average agreement, and 50% lacked agreement (Table 7).

#### 3.2.7. Welfare Assessment

Agreed definition: assessment of quality of life for individual horses; this should include regular monitoring of the training and competition management, stable and environmental management, health and veterinary management, nutrition and behavioural interactions, and the mental wellbeing of horses across all aspects of home and competition environments.

Welfare assessment was not agreed as a core domain; however, 57% of sub-domains within this area were agreed by the experts to be ‘essential’ in the management of sporthorse health and welfare, while 43% lacked agreement (Table 8).

#### 3.2.8. Measuring and Monitoring Sporthorse Health and Welfare

Overall, the expert panel rated 58 areas related to health and welfare monitoring as ‘*essential*’ (>70% agreement) to sporthorse management (Appendix A). Although there was no agreement on which measures are currently available to accurately assess sporthorse health and welfare, a large range of topics were identified across participants for what should be monitored to ensure sporthorse health and welfare. Key areas linked to sporthorse health and welfare management that the experts agreed should be monitored included health (2 measures), injury (11 measures), illness (1 measure), stable management (6 measures), fitness—workload—recovery (11 measures), behaviour (4 measures), and regular record keeping (7 measures) (Appendix A).

Poor agreement existed across experts for which factors should be assessed regularly and time scales for these (Appendix A). Of the 34 measures assessed, experts most frequently selected that daily monitoring should occur for 14 measures, weekly assessment for 2 measures, monthly assessment for 7 measures, and ad hoc assessment for 10 measures. However, consensus (>70% agreement) only occurred for daily assessment of horse recovery after exercise. Often, topics were rated as essential to monitor, but there was average to poor agreement that measures or tools currently exist to do this. For example, 80% rated monitoring surfaces as essential, but only 55% felt a suitable measure was currently available for this purpose; 100% rated monitoring of horse/rider interaction as essential, but only 48% felt a suitable measure or assessment method was currently available to do this.

Overall, 50% of the 12 identified areas for environmental and climate monitoring were defined as useful to monitor for sporthorse health and welfare, but respondents felt there are existing methods of assessment for only 17% of these areas. For stable management, 52% of the areas were identified as useful to monitor, but respondents felt methods to assess them currently existed for only 28% of these areas. For health and veterinary monitoring, 48% of the topics were found to be important to monitor regularly, but only 14% of respondents knew existing methods to assess them accurately. Surprisingly, for gait assessment monitoring, while 54% of the topics were reported to be useful for monitoring by the experts, no existing methods were rated as being currently available to assess them accurately. For fitness and recovery monitoring, 55% of the topics were reported to be useful to monitor, but only 10% of these felt known methods were available to assess them accurately. Similarly, for behaviour and welfare assessment monitoring, 50% and 40%, respectively, of the topics were identified as useful for monitoring; but again, respondents felt no methods were currently available that could accurately assess these areas. Within training monitoring, 43% of topics were selected as useful to monitor the health and welfare of sporthorses, but only 9% of respondents believed there were existing methods available to do this accurately. For competition monitoring, 62% of the areas were selected as useful to monitor, but only 32% of respondents felt there were existing methods to assess these accurately. These findings reveal large gaps between stakeholder perceptions for areas that require monitoring within sporthorse health and welfare and knowledge of currently available methods that can facilitate assessment accurately.

#### 3.2.9. Differences in Ratings across Participant Roles

No differences were found for agreement of the core domains between the levels of agreement between the five broad roles of participants defined in this study. However, respondents employed by national federations tended to rate sub-domains in core areas lower (less essential) than other participants. When evaluating the topics within the core domains, allied professionals, welfare experts, and veterinarians rated several subdomains significantly higher (more essential) for sporthorse health and welfare than equestrian federation employees (Table 9).

No significant differences in expert ratings were found between countries (*p* > 0.05). However, frequency analysis identified that specific examples of individual experts rating areas less than four (i.e., rating a specific domain or subdomain ‘*as not essential*’ for sporthorse health and welfare) were recorded across all participating countries (range: 33–100%).

#### 3.2.10. Participant Views on the Future of Equestrian Sports

Just under half (49%) of the participants in Stage Four felt that the future of equestrian sports was under threat, 33% thought it might be under threat, while 18% did not feel this way. Two core themes emerged that the participants felt underpinned the future of equestrian sports: (1) the future is uncertain and (2) there is poor awareness across the equestrian industry on equine welfare.

Participants stated the study results could be used to support the equestrian community by (1) helping to inform and support practice (through education and guidance), (2) to increase awareness (dissemination and showcasing of what is conducted well), and (3) to generate evidence (inform best practice and debate) (Table 10).

Participants also felt there were four key opportunities that equestrian sport could embrace to proactively improve sporthorse health and welfare: (1) credibility; (2) education; (3) regulation; and (4) showcasing best practice (Table 11).

While the participants felt there was a need for increased knowledge and education on how to manage sporthorse health and welfare, they also felt there was a need to generate change and increase awareness particularly targeted at riders from a young age. They also felt that there was a need to manage the demands of competition, with riders understanding they should manage horses as animals and partners not as machines, and to manage the horses’ environment accordingly. Participants also felt there was a need for course designers, competition organisers, and governing bodies to control performance and technical demands so that expectations remain realistic and avoid increasing the demands to such a degree that there is a risk of having a negative impact on sporthorse health and welfare.

## 4. Discussion

This is the first consensus report summarizing international equestrian expert opinion on which factors are essential within the contemporary management of sporthorse health and welfare.

Across all results, individual participants are consistent in how they rated whether a domain or sub-domain was essential, or not, for sporthorse health and welfare across the four rounds of the study, i.e., they did not change their mind as the project progressed. However, opinions between individual participants regarding which core domains and sub-domains were essential for sporthorse health and welfare were more varied, representing the breadth of diverse opinions across the group (and the equestrian industry).

Participants reached consensus (agreement) that the core domains training management, competition management, young horse management, health status and veterinary management, and the horse–human relationship were essential within the management of sporthorse health and welfare. Participants did not reach consensus that stable and environmental management, and welfare assessment were essential components of sporthorse health and welfare management. On further questioning, the majority of participants felt that these areas ‘*were not essential*’ as they were already managed well.

While all experts agreed that the five core domains were ‘*essential*’ for the management of sporthorse health and welfare, there was little consensus on which areas or sub-domains underpinning these core domains were essential for the contemporary management of sporthorse health and welfare. Across all areas, 30% of subdomains were agreed to be ‘*essential*’ (8–42% range within specific domains); agreement for the sub-domains that underpinned the core domain training and competition management recorded the least agreement by participants. In addition, there was poor agreement for which areas should be monitored regularly to assess sporthorse health and welfare, and participants felt few effective tools were currently available to do this.

The results showcase the diversity of opinion present across individual equestrian stakeholders and highlight the need for clear, agreed, and evidence-informed industry guidelines to support all levels of equestrian stakeholders to manage their horse’s health and welfare optimally. Equestrianism has a strong traditional base, with many management decisions underpinned by historic practice [15,47] or individuals’ experiences [8]; this can result in a dogmatic stance where existing anecdotal practice is continually reinforced to be the best approach, rather than being questioned. The broad range of expert opinions reported, the dichotomies observed between rating areas that underpin core domains as not essential, and the general lack of knowledge of monitoring tools to assess sporthorse health and welfare suggest that an element of this exists within international equestrian stakeholders.

While the application of science into practice and the uptake of evidence-informed approaches were advocated across the management of sporthorse [3,8], these results suggest that there are barriers to implementation. The challenge facing horse sports combines a lack of empirical evidence to underpin practice with stakeholders who can struggle to keep themselves updated on emerging research and technology, either due to a lack of time or access to the ‘best’ information sources. Effective scientific communication to relevant audiences is acknowledged as challenging not just in the equestrian sphere [48]. This, combined with a general lack of evidence-based research to inform practice in many of the areas highlighted as essential across core domains and sub-domains, indicates areas requiring education and further research. This sentiment was echoed by our experts, who articulated a need for greater education and engagement to generate change and promote evidence-informed approaches to sporthorse health and welfare. The use of evidence syntheses or systematic reviews is widespread in human medicine to underpin evidence-based approaches [49] and may be a viable starting point for further evaluation of topics to provide evidence-informed approaches to common elements of sporthorse health and welfare management.

### 4.1. Essential Versus Important: Evidence-Informed or Opinion-Based?

While generally the median ratings indicate that the majority of participants rated the core domains (range: 7–9) and sub-domains as essential to the management of sporthorse health and welfare, individual variability in grading was a common theme across all stages of the Delphi process (minimum rating range recorded: 0 to 5; maximum rating range recorded: 0 to 9). For example, young horse management, which achieved the highest consensus rating, was still rated at 0 by some participants.

An individual’s life experience contributes to their perception of situational awareness and their own knowledge base [50,51]. Generally, individuals with more years of experience within a specific sector or role are believed to have accumulated knowledge and skills in that field and therefore deliver high-quality care. However, interestingly, empirical evidence suggests an inverse relationship exists between the number of years physicians were in a role and the quality of care they provide [52]. Definitions for what constitutes experience are heterogeneous across the literature and are often related to frequency of exposure, the severity of situations experienced, or an individual’s indirect experiences [53]. However, knowledge is not just based on experience; for example, there is a difference between riding a horse in a warm-up at a competition and watching a warm-up on television. An individual’s education, personal research, governing body guidance, and information from peers and public sources can all influence ‘knowledge’ [53]. Therefore, it is essential to distinguish between an individual’s self-assessed knowledge and his/her scientific knowledge of the area under evaluation, and to determine whether these match and what influence each has on decision-making when considering the judgements made [53]. Consequently, to evaluate if individual expert experience could be influential, additional analyses were undertaken to compare ratings for the core domains across the participating equestrian nations and expert roles.

No significant differences were found in expert ratings between countries; however, across all countries, there were examples of individual experts rating variable domains and subdomains ‘as not essential’ for sporthorse health and welfare. These results suggest that a dichotomy exists across stakeholders at the highest level of equestrian sport as to which factors are truly essential to underpin contemporary management of the elite equine athlete, rather than being associated with how advanced equestrian practice is within an individual nation. This also implies that although in practice it is sometimes assumed that some countries have, in general, a more up-to-date or accurate view on sporthorse health and welfare, this does not seem to be the case consistently within the results of this Delphi study.

Interestingly, no differences in agreement for how essential the core domains were within sporthorse health and welfare management existed between the five broad roles of participants defined in this study. However, there was a trend in some subdomains, outlined in Table 9, for respondents employed by national federations to rate these lower (less essential) than other participants. These differences may be associated with expert roles and the focus and depth of training or experience within these groups across the subdomains where differences exist. Alternatively, these could be associated with cultural variances, with traditional practices informing different priorities within sporthorse management. The trend observed aligns with experts’ perspectives that there is poor awareness across the equestrian industry of equine welfare and their recommendation that the results of the Delphi should be used to increase awareness of, and showcase, good welfare.

Interestingly, the results of two large-scale surveys of the equestrian and general public in 2022, by the FEI Equine Ethics and Wellbeing Commission (EEWC), identified that 67% of the public and 78% of equestrian stakeholders felt that existing welfare standards in horse sports required improvement [54,55]. The Delphi results indicate that variable consensus exists across many areas that impact the management of sporthorse health and welfare and indicate a consistent approach to the assessment of sporthorse management is needed across global stakeholders. Further work should also be undertaken to ensure parity in sporthorse management that prioritizes health and welfare outcomes as well as performance; the consensus presented here identifies five agreed core domains that could provide a starting point to develop guidance that could be used to underpin this.

A good starting point to promote sporthorse health and welfare and establish an effective SLO for horse sports would be to develop an evidence-based infrastructure to inform the management, training, and riding practices utilized [2,11]. The core domains identified in this study could provide a viable starting point to develop this. The development and recently tested ethical framework from Campbell [1] is another example of a framework that can provide information to assist stakeholders in making contextual decisions on ethical questions related to equestrian sports. However, as shown in this study and the results of the FEI EEWC survey results [54], similar to other sports and animal welfare sectors, individual stakeholders will have different perspectives as to what is an ethical practice depending on their personal background, moral viewpoint, and on the ethical theory they adopt. The recognition that conflicts will occur, and documenting these is an integral and necessary part of future developments to ensure that the broader equestrian community engage with ethical frameworks or advancements relating to sporthorse health and welfare [1].

### 4.2. Management of Core Domains

Although five core areas were rated as essential, generally, there was a lack of agreement as to which sub-topics within these areas were essential to manage sporthorse health and welfare. In some instances, for example, for welfare assessment, sub-domain topics were rated as essential where the core domain they related to was not. In others, such as training management and horse–human relationship, career longevity and the rider/groom/coach experience were rated as more essential, respectively, than areas which logically underpin these areas that were not felt to be essential, e.g., training program and fitness, and relationship with the horse. Caution needs to be applied when postulating reasons for this potential disconnect, as the Delphi does not allow for in-depth understanding of participant interpretation of ‘essential’ versus ‘important’.

In response to the variability of rating observed, participants were asked their opinions as to why there was a trend for some individuals to rate some areas, such as welfare, as not essential within sporthorse management. Generally, participants believed that lower ratings were associated with areas that already benefitted from being well managed or that were subject to regulatory frameworks, making them not essential to be addressed currently. Assessing our own confidence in the decisions we make, as well as the evidence available is purported to be influential when making adaptive decisions [56]. A regulatory framework is defined as a particular set of rules, ideas, or beliefs that are used in order to deal with problems or to decide what to do [57]. The presence of a regulatory framework within a sector infers confidence in users that a model is in place to help control risk, in essence creating the infrastructure to underpin social license and give the public confidence that an area is being managed well. This concept could partially explain why some experts here felt domains were not essential due to their perception that regulatory frameworks to manage elements of sporthorse health and welfare are in place. However, most of the regulatory frameworks within equestrian sports were developed by national federations or are discipline specific and largely based on anecdotal rather than evidence-based information [1]. Caution is necessary, therefore, when equating the presence of a framework to the highest standards for what the framework encapsulates, and a judgment that a framework equals a well-managed area. The presence of a framework could translate over time to a sense of complacency that an area is already well managed, resulting in a lack of scrutiny and the efficacy of the framework may not then be questioned. It should also be noted that within SLO, it is the public’s perception of how sporthorse health and welfare are managed that are key to secure the future of horse sports. To showcase to the public, but also improve welfare internally in the equestrian industry, it is important that stakeholders can identify which areas are important within a framework for sporthorse health and welfare [1,12]. Currently, a paucity of empirical data exists to evaluate how the areas within the domains and sub-domains are being managed in practice and studies to generate this evidence should be prioritized to support future assessment tools.

The lack of agreement for overarching assessment tools combined with the low level of agreement within sub-domains that should support the essential core domains is also concerning. Questions from the public regarding the use of horses in competitive sport require stakeholders to be well informed and able to make contextual decisions about what is or is not an appropriate response in specific situations and being able to demonstrate that their actions are evidence informed and meet the needs of sporthorses [1,3,58]. A fundamental knowledge and understanding of core concepts that underpin sporthorse management, such as exercise physiology, behavior, principles of training, biomechanics, and nutrition are needed to manage sporthorses effectively [8]. The results here suggest further education is required to ensure this foundation knowledge is in place across all equestrian stakeholders, even at this top level of the sport. Our results also reflect the lack of an agreed consensus and subsequent framework for what constitutes optimal sporthorse management across the industry. Some good examples exist, for example, the Swedish Animal Welfare Act 2018, which includes regulation that all horses should receive daily turnout opportunities [59], and Tierschutz im Pferdesport (‘Animal Protection in Equestrian Sport’) [60] published by the German Equestrian Federation after consultation with the German Agricultural Ministry and other equestrian stakeholders as a guide to promote more ethical management and training of sporthorses [61]. However, generally, management guidance to date was developed *ad hoc* or for specific discipline or user groups, reducing their transferability [1,6] and more work is needed in this area [62].

Campbell et al. [1] advocates the development of a broader ethical framework for sporthorses building on Mellor’s five domains model [63] within the context of the sporthorse disciplines. In essence, this approach generates a harm–benefit analysis that integrates central principles: reducing negative welfare and maximizing positive welfare effects to give horses a life worth living, identifying, and mitigating against avoidable and unnecessary risk, and complying with governing body regulations and legislation. Building on the areas where consensus is agreed and not agreed here, combined with a program of education and research, this approach could develop to showcase how the industry’s duty of care to the sporthorse is being managed [3,8]. The development and testing of an overarching framework as proposed by Campbell et al. [1] could also be useful for group decision making on a wide variety of ethical questions [64] and could help decrease existing gaps in the equestrian industry between identified essential domains for sporthorse health and welfare and the lack of supporting sub-domains and tools evaluating them. It is also critical to understand the motivators of human behavior when evaluating how different stakeholders influence sporthorse management, including veterinarians and officials [16]. Moving forward this approach could ultimately provide evidence-informed justification for the use of horses in sport by demonstrating their experience is overall a positive one [54,55,65,66,67].

### 4.3. Prioritizing Performance and Welfare

Across the Delphi exercise, areas associated with maintaining the ability of the sporthorse to perform a job were rated as more essential than factors linked to career longevity, reducing injury, and optimizing horse welfare. Similarly, areas linked to long-term health and monitoring welfare, in particular mental wellbeing, and quality of life, including equine behavior, social interaction, and opportunities for free movement, were consistently rated lower in terms of being essential. This perspective is perhaps indicative of the trade-off between the utility of a sporthorse and its welfare; those involved in sporthorse management may more clearly observe a tangible link between improving a horse’s physical health and welfare, and superior performance. However, recognition of the positive impact of improved mental wellbeing on performance is less commonplace across equestrian sport. Further research is needed to clearly articulate how mental state, not just physicality, contributes to performance, and to propose user-friendly methods to measure the impact of providing horses with good welfare rather than setting the bar at meeting their essential needs.

As indicated in these results and from recent surveys [1,64,68], this perspective is not universal across equestrianism and does not reflect the views of the wider public [54,55]. However, it does highlight the fact that different equestrian and/or non-equestrian communities have different views on equine welfare. This includes which facets of management should be prioritized and what is ethically allowed based on their moral viewpoint, influenced by past personal and professional experience, and judgment. Applying or developing additional frameworks such as the five domains model [63] or the equine focused AWIN [69] could contribute to the assessment of equine welfare and identification of specific sporthorse welfare indicators. These could be applied across global stakeholders and within multiple contexts to promote a positive welfare approach in the management of sporthorses.

Most respondents of the EEWC survey in 2022 indicated that they believed that horses would only continue to be involved in sport if equine welfare is improved [54]. The Delphi results identified that stakeholder perception of how horse welfare is perceived and assessed is variable, with some believing welfare is assessed well and others identifying this as an area requiring improvement. Effective governance is key to embedding a global infrastructure that promotes sporthorse health and welfare, and which recognizes that optimum equine performance is underpinned by good health and a good quality of life. The FEI’s EEWC outline their vision for the future across their 24 recommendations: A ‘Good life for horses’ advocating an ethical and evidence-based approach that establishes a trusted and proactive culture of accountability, responsibility, and transparency across equestrian sport [70]. Increased research and education are key enablers identified to support this vision to inform practice and regulation across the sector, underpin decision-making to promote a good life for horses, enable accurate welfare assessment, and support policy makers and regulatory bodies to act as an advocate for the horse across competitive horse sports [2,3,55]. A good starting point, suggested by Douglas et al. [3], is for individuals to embrace their collective responsibility by asking “*Should I*?” rather than “*Can I*?” when considering sporthorse training and management practices to demonstrate a personal commitment to prioritizing horse welfare [71].

### 4.4. Assessing Sporthorse Welfare

A key outcome of the Delphi was the lack of consensus that welfare assessment should be considered a core domain of sporthorse management. Interestingly, welfare assessment in training, equine quality of life, suitability of training and the environment, and post-career management were all highly rated as essential areas individually, but these topics did not score highly within related core domains, e.g., training management, and competition management. These results again suggest a disconnect in how experts value welfare assessment across different contexts. When questioned, participants stated the future of equestrian sports was uncertain and they felt that there is a poor awareness of equine welfare across the equestrian industry. The outcomes presented here suggest sporthorse welfare is a key concern to national and international equestrian stakeholders, but the context for how and where it is assessed is more difficult to ascertain consistently. This highlights an interesting dilemma as the public and broader equestrian population feel competitive sporthorse welfare is currently lacking and not prioritized [54]. The experts voiced a need for increased credibility for individuals and national and international federations with regards to equine welfare. They want more education and guidance to inform practice, monitoring, and decision-making, with increased regulation and acknowledgement of existing good practice. The dichotomy that exists across stakeholders valuing the importance of equine welfare but then not rating it as a core domain and proposing that further education and guidance is needed, suggests welfare is poorly understood, perhaps because it is a complex multifactorial concept. The results here suggest health and welfare can be interchangeable concepts when evaluating equine quality of life, and there is a need to consider how welfare assessment, embracing the five domain model approach, contributes to a horse having a good life [72]. There appears to be an opportunity for key global stakeholders to work together to support their members and showcase to the public that horse welfare is being prioritized, which is recognized in the FEI EEWC’s recommendations [70]. The challenge to bring together a diverse international community such as exists across equestrian sports and achieve a unified perspective and adoption of shared welfare guidelines should not be underestimated. It is unlikely that one simple solution exists to achieve this, and different perspectives and actions will need to be implemented to achieve success and upskill different audiences, under the leadership and direction of global regulatory stakeholders.

Assessing animal welfare is a complex topic [73] and it is acknowledged that additional positive indicators of welfare and emotions are needed across all species and not just for the horse [74,75]. There is a growing body of evidence that expression of behavioral diversity, while not validated as a positive welfare indicator, may be an important component of a welfare assessment tool [74,76]. The results here highlight that, with the exception of within young horse management, monitoring of behavior, opportunities for free activity not related to training, and social interaction were not considered essential for the management of sporthorse welfare. As highlighted previously, tools do exist to assess equine welfare; however, these are not contextualized to the management of the sporthorse, during training or competition. Given the poor perception of sporthorse welfare and welfare regulation in horse sports by the public and general equestrians, consideration should be given to prioritizing the development of positive welfare indicators that could be adopted across sporthorse disciplines to enable more effective monitoring of sporthorse welfare.

One of the main challenges in assessing sporthorse welfare is the potential conflict between the demands of training and competition, and how the basic needs of the horse are met [68]. Sporthorses are acknowledged to have some sense of control over their own actions and behavior; this concept is expanded by Holt et al. [77], who argue that sporthorses should have athlete status in their own right. The concept of a horse as an athlete raises some interesting questions. Stress related to training, competition, injury, and lifestyle choices, is an inherent component of elite sport, which human athletes consciously embrace [78]. For the equine athlete, engagement is not entirely voluntary, and while superficially it may appear to the public that horses can be made to engage in competitive activities, in reality, only horses that are willing to perform will become elite sport horses. However, as a result of participation in sport, horses, as with all athletes, are likely to experience transient periods of physiological and psychological distress and are exposed to the potential for injury and possibly fatality. This can lead to a disconnect between the horse’s ‘best’ life and life as an athlete [68]. Bearing this in mind, it may be prudent to assess sporthorse quality of life as a continuum rather than engaging in episodic welfare assessments that do not consider the full repertoire of activities, behaviors, and environments that the sporthorse encounters; an approach advocated by Mellor [72]. Developing such an approach would be beneficial and could be utilized to justify the ongoing use of the horse in sport if the horse’s quality of life overall can be evidenced to be positive.

### 4.5. Study Limitations

The outcome of Delphi studies is driven by the experts who actively participate within it. Stringent inclusion criteria were applied to attempt to ensure selected participants represented a national and international equestrian perspective; however, it should be noted that if participants did not raise an area throughout the process, it was not included. Analogous to prior Delphi studies, attrition was observed between rounds, however, at all stages, participation exceeded accepted guidelines for the minimum number of participants needed to achieve valid consensus within a targeted discipline area. It was beyond the scope of the Delphi to evaluate why participants rated the areas and topics as they did. Where ratings of core domains did not achieve consensus, participants were asked why they thought this was but not everyone responded. This could be related to individual interpretation of the difference between essential and important; the results suggest this cohort often determined an area to be important but not essential where they felt some formal framework, regulation, or guidance already existed. It could also be that repetition of subdomains across domains influenced participant rating, for example, rating subdomains consistently regardless of which domain they are attached to, or conversely, could potentially feel a subdomain is already rated highly and rate it lower on subsequent presentation. Understanding differences in stakeholder use and interpretation of language should be considered in future studies to ensure accuracy of grading and to promote inclusivity. Initial consultation in Stage 1 was in English and feedback identified that English was not their first language for all stakeholders, and this possibly influenced participant interpretation of questions. Therefore, for subsequent rounds, questions were available in French, Dutch, Spanish, and German with responses translated to English prior to analysis. Further work applying qualitative methodologies to delve deeper into equestrian stakeholder perception and gain a greater understanding of factors that influence decision-making related to the management of sporthorse health and welfare is also warranted.

### 4.6. Recommendations

The findings of this study identify areas and topics in which expert national and international equestrian industry stakeholders have an agreed consensus, no consensus, or limited consensus, as to whether they are essential to manage sporthorse health and welfare or not. This is the first in-depth view of the national and international equestrian industry’s opinion and feelings on sporthorse health and welfare; it provides a starting point for debate and identifies areas that should be prioritized moving forward.

The results identify domains and sub-domains where the elite equestrian industry has no consensus as to whether these areas are essential to the management of sporthorse health and welfare or not. A logical next step is to investigate whether these areas can be underpinned by scientific literature or should be prioritized for scientific investigation soon to evaluate whether they are essential for sporthorse health and welfare.

Based on our results, we propose the following recommendations to be actioned:Evaluate the opinions of other equestrians (non-elite) and the wider public on these areas for comparison with the current findings to provide a broader insight regarding the range of opinions on sporthorse welfare. This might help to identify and prioritize areas as more or less urgent to investigate and/or optimize in the future.Gather empirical evidence to understand what practices/management are being implemented across different countries, disciplines, competitions, and individuals. Increased research undertaken in partnership with industry is required to generate data to support evidence-informed practice. Increased monitoring, record keeping, and research will enable good practice to be identified and showcased to the wider equestrian communities and the public to generate a culture in which quality of life for sporthorses comes first.Encourage national and international federations to provide targeted education and guidance, policy development, and regulation to improve the management of sporthorse health and welfare. Specifically, there is a need for increased education to improve understanding of what is welfare, how to assess it, and how it can enhance equine performance across equestrian stakeholders. Development of effective dissemination strategies for education, tools, and guidance should also be adapted for different community needs.Using the consensus agreed here, there is an opportunity for core stakeholders to come together and accelerate change to promote good practice through the development of a sporthorse welfare charter and production of evidence-informed guidelines to support the management and monitoring of these areas.

## 5. Conclusions

After the four rounds of Delphi consultation, global equestrian experts agreed that five core domains were essential within the management of sporthorse health and welfare: ‘training management’, ‘competition management’, ‘young horse management’, ‘health status and veterinary management’, and the ‘horse–human relationship’. Two further domains, ‘stable and environmental management’ and ‘welfare assessment’ were not agreed to be essential components within sporthorse health and welfare management, but this was largely because most respondents felt that these areas were already managed well. Although five core areas were rated as essential, generally, there was a lack of agreement for which sub-topics within these areas were essential to manage sporthorse health and welfare.

Individual expert opinion was consistent across the rounds of the Delphi, but varied widely between participants reflecting a disconnect across equestrian stakeholders, which was not related to their role or country of origin. Participants stated the future of equestrian sports was uncertain and felt that there is poor awareness across the equestrian industry for how to assess and manage equine welfare. Areas associated with maintaining the ability of the sporthorse to perform a job were generally rated as more essential than factors linked to career longevity, reducing injury and welfare. Factors linked to long term health and monitoring welfare, in particular mental wellbeing, and quality of life, including equine behavior, social interaction, and opportunities for free movement, were consistently rated lower in terms of being essential.

The results suggest that increased education and guidance, and further policy development and regulation, combined with research to inform practice alongside gaining a greater understanding of how perspectives and knowledge varies across equestrian communities, are needed to be able to support stakeholders to optimize sporthorse management. The development of a sporthorse welfare charter and production of evidence-informed guidelines to support the management and monitoring of sporthorses’ health and welfare are recommended. Proactive engagement across all levels of horse sport is needed to establish and maintain equestrianism’s social license and to showcase to the public how the physical and psychological needs of sporthorses are managed to provide horses with a good life and to safeguard the future of equestrian sports.

## Figures and Tables

**Figure 1 animals-13-03404-f001:**
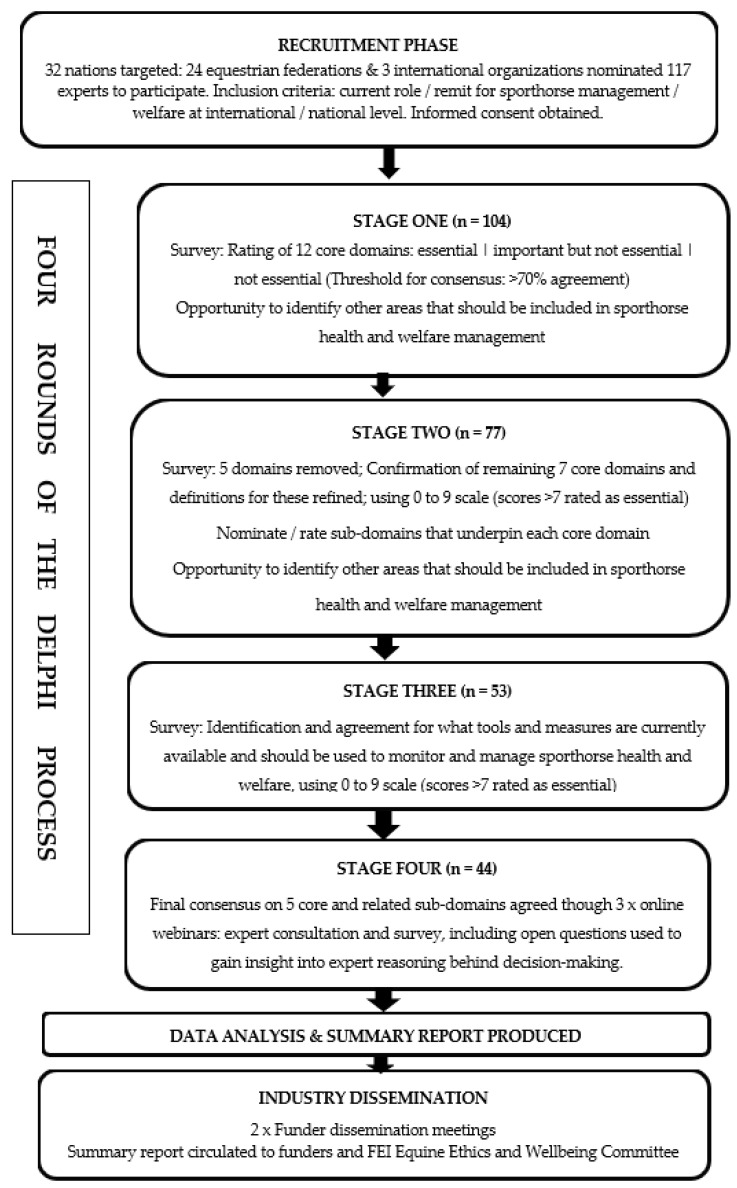
Flow chart of the Delphi process.

**Figure 2 animals-13-03404-f002:**
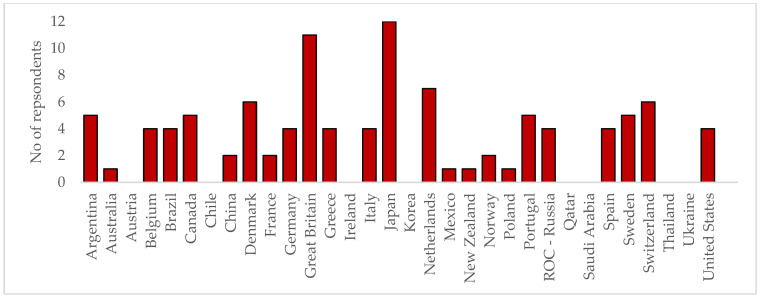
Equestrian federations that participated in Stage One of the Delphi study.

**Figure 3 animals-13-03404-f003:**
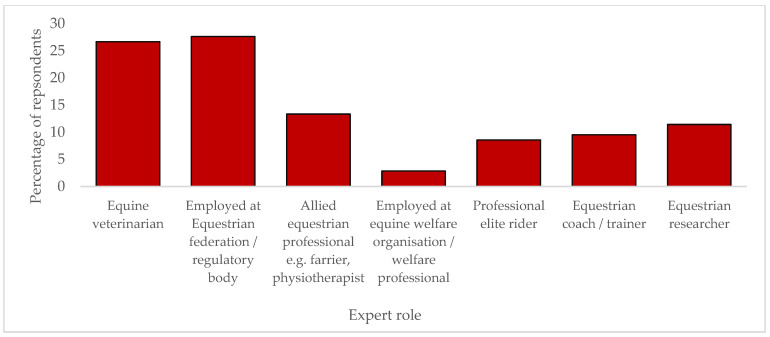
Distribution of expert roles across the Delphi study.

**Table 1 animals-13-03404-t001:** Overview of agreement across core domains (green: agreement that the area is essential; yellow: above average agreement that the area is essential; and red: no agreement that the area is essential for sporthorse health and welfare). CVR: content validity ratio; CVI: content validity index.

Core Domains:	Agreement (CVR): Essential to Sporthorse Health and Welfare	Range of Scores: Median (Range) (0 Not Essential to 9 Essential)
Young horse management	0.89	8 (0–9)
Training management	0.78	8 (2–9)
Health status and veterinary management	0.78	8 (4–9)
Horse-human relationship	0.77	8 (1–9)
Competition management	0.69	8 (4–9)
Stable and environment management	0.49	8 (1–9)
Welfare assessment	0.35	8 (1–9)
Areas rejected as distinct domains in Stage 1 *:	**Nutrition** **Behaviour** Use of allied professionals **Biomechanical/locomotion assessment** Judges, officials, and rules	
Consistency of agreement (Cronbach’s Alpha):	0.85 (0.80–0.86)	
Average agreement (CVI):	0.68	

* Areas highlighted in bold were subsequently integrated into the remaining areas based on expert feedback.

**Table 2 animals-13-03404-t002:** Overview of agreement: training management (green: agreement that the area is essential; yellow: above average agreement area that the area is essential; and red: no agreement that the area is essential for sporthorse health and welfare); CVR: content validity ratio; and CVI: content validity index.

Training Management Sub-Domains:	Agreement (CVR): Essential to Sporthorse Health and Welfare	Range of Scores: Median (Range) (0 Not Essential to 9 Essential)
Monitoring career longevity	0.78	8 (0–9)
Training environment	0.75	9 (2–9)
Physical workload	0.69	8 (0–9)
Tack and equipment	0.69	8 (2–9)
Readiness for work	0.69	8 (4–9)
Fatigue	0.66	8 (3–9)
Rehabilitation recovery	0.66	8 (3–9)
Fitness	0.63	8 (3–9)
Psychological workload	0.60	8 (3–9)
Competition frequency	0.60	8 (5–9)
Behaviour	0.60	8 (5–9)
Nutrition	0.60	7 (5–9)
Recovery	0.54	8 (5–9)
Training programme	0.51	8 (3–9)
Climate management	0.51	7 (5–9)
Social interaction and free movement	0.42	7 (4–9)
Career longevity	0.42	7 (1–9)
Monitoring of training	0.29	8 (3–9)
Areas rejected for inclusion in Stage 1:	Injury history Managing horse’s body temperature Lameness/gait assessment Use of supplements Thermoregulatory management	
Consistency of agreement (Cronbach’s Alpha):	0.93 (0.92–0.93)	
Average agreement (CVI):	0.59	

**Table 3 animals-13-03404-t003:** Overview of agreement: competition management (green: agreement that the area is essential, yellow: above average agreement that the area is essential, and red: no agreement that the area is essential for sporthorse health and welfare); CVR: content validity ratio; and CVI: content validity index.

Competition Management Sub-Domains:	Agreement (CVR): Essential to Sporthorse Health and Welfare	Range of Scores: Median (Range) (0 Not Essential to 9 Essential)
Health monitoring of competition horses	0.75	8 (4–9)
Competition performance management	0.68	8 (5–9)
Competition frequency	0.68	8 (2–9)
Travel management	0.68	8 (3–9)
Nutrition during competition	0.68	8 (1–9)
Behaviour during competition	0.59	8 (1–9)
Rules, officials, and regulations	0.52	8 (4–9)
Management of body temperature	0.43	8 (1–9)
Monitoring of incidents during competition	0.40	8 (4–9)
Performance in competition	0.37	7 (5–9)
Competition environment and infrastructure	0.37	8 (2–9)
Social interaction and free movement	0.27	7.5 (2–9)
Areas rejected for inclusion in Stage 1:	Thermoregulatory management Injury history	
Consistency of agreement (Cronbach’s Alpha):	0.92 (0.90–0.92)	
Average agreement (CVI):	0.53	

**Table 4 animals-13-03404-t004:** Overview of agreement: young horse management (green: agreement that the area is essential, yellow: above average agreement that the area is essential, and red: no agreement that the area is essential for sporthorse health and welfare); CVR: content validity ratio; and CVI: content validity index.

Young Horse Management Sub-Domains:	Agreement (CVR): Essential to Sporthorse Health and Welfare	Range of Scores: Median (Range) (0 Not Essential to 9 Essential)
Pre-conditioning work	0.84	8 (3–9)
Monitoring of training and management of the young horse	0.81	8 (4–9)
Provision of social interaction	0.72	8 (4–9)
Provision of appropriate free exercise	0.72	8 (4–9)
Behaviour of the young horse	0.69	8 (4–9)
Youngstock management	0.66	8 (2–9)
Young horse competitions	0.66	8 (0–9)
Nutrition for the young horse	0.59	8 (3–9)
Tack and equipment for the young horse	0.59	8 (4–9)
Breeding sustainability	0.50	8 (2–9)
Training of the young horse	0.31	7 (6–9)
Physical and mental assessment of ridden young horses	0.25	7 (2–9)
Breeding	0.16	7 (0–9)
Areas rejected for inclusion in Stage 1:	Assessment of conformation	
Consistency of agreement (Cronbach’s Alpha):	0.92 (0.92–0.93)	
Average agreement (CVI):	0.58	

**Table 5 animals-13-03404-t005:** Overview of agreement: health status and veterinary management; (green: agreement that the area is essential; yellow: above average agreement that the area is essential; and red: no agreement that the area is essential for sporthorse health and welfare); CVR: content validity ratio; and CVI: content validity index.

Health Status and Veterinary Management Sub-Domains:	Agreement (CVR): Essential to Sporthorse Health and Welfare	Range of Scores: Median (Range) (0 Not Essential to 9 Essential)
Injury management	0.81	8 (3–9)
Orthopaedic health	0.78	8 (3–9)
Hoof management	0.78	8 (5–9)
Movement and lameness assessment	0.78	8 (1–9)
Respiratory health	0.75	8 (5–9)
Gastro-intestinal health	0.68	8 (5–9)
Preventative healthcare	0.68	8 (5–9)
Medication monitoring	0.56	8 (4–9)
Health monitoring	0.56	8 (1–9)
Health and veterinary management monitoring	0.55	8 (3–9)
Cardiovascular health	0.46	8 (4–9)
Conformation	0.41	7 (2–9)
Areas rejected for inclusion in Stage 1:	Nutritional management Monitoring of use of allied professionals	
Consistency of agreement:	0.92 (0.92–0.93)	
Average agreement (CVI):	0.65	

**Table 6 animals-13-03404-t006:** Overview of agreement: horse–human relationship; (green: agreement that the area is essential; yellow: above average agreement that the area is essential; and red: no agreement that the area is essential for sporthorse health and welfare); CVR: content validity ratio; and CVI: content validity index.

Horse–Human Relationship Sub-Domains:	Agreement (CVR): Essential to Sporthorse Health and Welfare	Range of Scores: Median (Range) (0 Not Essential to 9 Essential)
Rider experience	0.94	8 (5–9)
Assessment of horse–human interactions	0.81	8 (1–9)
Groom experience	0.81	8 (1–9)
Coach/trainer experience	0.77	7 (5–9)
Professional body–horse relationship	0.52	8 (1–9)
Rider–horse relationship	0.35	8 (1–9)
Groom–horse relationship	0.35	8 (1–9)
Equestrian federation–horse relationship	0.35	8 (3–9)
Coach/trainer–horse relationship	0.26	8 (0–9)
Assessment of horse–horse interactions	0.26	7 (1–9)
Rider skill and fitness	0.19	7 (4–9)
Areas rejected for inclusion in Stage 1:	None rejected	
Consistency of agreement (Cronbach’s alpha):	0.93 (0.92–0.93)	
Average agreement (CVI):	0.51	

**Table 7 animals-13-03404-t007:** Overview of agreement: stable and environment management; (green: agreement that the area is essential; yellow: above average agreement that the area is essential; and red: no agreement that the area is essential for sporthorse health and welfare); CVR: content validity ratio; and CVI: content validity index.

Stable and Environment Management Sub-Domains:	Agreement (CVR): Essential to Sporthorse Health and Welfare	Range of Scores: Median (Range) (0 Not Essential to 9 Essential)
Horses’ behavioural needs	0.78	8 (4–9)
Sustainability footprint	0.78	8 (0–9)
Monitoring of stable and environment management	0.63	8 (0–9)
Rules and regulations for stable management	0.53	8 (4–9)
Environmental climate	0.47	8 (4–9)
Stable environment	0.44	8 (0–9)
Travel impact	0.40	8 (2–9)
Tack and equipment	0.05	7 (3–9)
Areas rejected for inclusion in Stage 1:	Turnout/opportunities for exercise not related to training or competition or being ridden Surface management Recording stereotypical behaviours	
Consistency of agreement (Cronbach’s alpha):	0.83 (0.79–0.83)	
Average agreement (CVI):	0.53	

**Table 8 animals-13-03404-t008:** Overview of agreement: welfare assessment; (green: agreement that the area is essential; red: no agreement that the area is essential for sporthorse health and welfare); CVR: content validity ratio; and CVI: content validity index.

Welfare Assessment Sub-Domains:	Agreement (CVR): Essential to Sporthorse Health and Welfare	Range of Scores: Median (Range) (0 Not Essential to 9 Essential)
Welfare assessment—training	0.81	8 (2–9)
Suitability of exercise and training programmes	0.77	8 (2–9)
Quality of life	0.77	8 (2–9)
Post career management	0.74	8 (2–9)
Welfare assessment management	0.68	8 (2–9)
Welfare assessment—competition	0.64	8 (5–9)
Suitability of tack and equipment	0.44	8 (1–9)
Areas rejected for inclusion in Stage 1:	Welfare assessment monitoring	
Consistency of agreement (Cronbach’s alpha):	0.93 (0.92–0.93)	
Average agreement (CVI):	0.69	

**Table 9 animals-13-03404-t009:** Subdomains within core domains where significant differences existed between expert ratings; KW: Kruskal–Wallis; and MWU: Mann–Whitney U.

Domain	Subdomain (*p* Value KW)	Expert Groups	*p* Value MWU
Training Management	Physical workload (*p* = 0.0160) Equine behaviour (*p* = 0.026)	Allied professionals rated higher than equestrian federation employees	*p* = 0.037 *p* = 0.028
	Fatigue assessment (*p* = 0.039) Training environment (*p* = 0.010)	Allied professionals rated higher than equestrian federation employees Welfare experts rated higher than equestrian federation employees	*p* = 0.012 *p* = 0.002 *p* = 0.013 *p* = 0.007
	Recovery (*p* = 0.003) Climate management (*p* = 0.026) Social interaction and free movement (*p* = 0.005) Career longevity (*p* = 0.044) Readiness for work (*p* = 0.007)	Welfare experts rated higher than equestrian federation employees	*p* = 0.013 *p* = 0.039 *p* = 0.001 *p* = 0.043 *p* < 0.001
Competition Management	Performance management (*p* = 0.05) Competition environment and infrastructure (*p* = 0.05)	Welfare experts rated higher than equestrian federation employees	*p* = 0.018 *p* = 0.012
Stable Management	Equine behavior assessment (*p* = 0.05)	Allied professionals rated higher than equestrian federation employees Welfare experts rated higher than equestrian federation employees	*p* = 0.01 *p* = 0.02
Young Horse Management	Pre-conditioning work (*p* = 0.004) Monitoring training (*p* = 0.003)	Allied professionals rated higher than equestrian federation employees Welfare experts rated higher than equestrian federation employees	*p* < 0.001 *p* < 0.001 *p* = 0.003 *p* = 0.003
	Monitoring nutrition (*p* = 0.011)	Welfare experts and veterinarians rated higher than equestrian federation employees Veterinarians rated higher than equestrian federation employees	*p* = 0.027 *p* = 0.008
	Free exercise opportunities (*p* = 0.006)	Allied professionals rated higher than equestrian federation employees	*p* = 0.014
Health Status and Veterinary Management	GI health (*p* = 0.017) Respiratory health (*p* = 0.013) Injury management (*p* = 0.19)	Welfare experts rated higher than equestrian federation employees	*p* = 0.033 *p* = 0.013 *p* = 0.029
	Movement and lameness assessment (*p* = 0.012)	Allied professionals rated higher than equestrian federation employees Welfare experts rated higher than equestrian federation employees	*p* = 0.004 *p* = 0.005
Human–Horse Relationship	Coach/trainer horse relationship (*p* = 0.006)	Veterinarians rated higher than equestrian federation employees	*p* = 0.050
Welfare Assessment	Equine quality of life (*p* = 0.013)	Welfare experts rated higher than equestrian federation employees	*p* = 0.02

**Table 10 animals-13-03404-t010:** Participant views on how the Delphi results should be used.

Help and inform practice (education)	Outline how to do the best for the horses Educate on scientific knowledge available Work with equestrian community to determine next steps Publish results: user friendly formats accessible to all stakeholders especially riders Develop into equine welfare guidance/strategy Seminars to support education across stakeholders Engage with animal activist sector—in an explanatory/scientific manner
Increase awareness (dissemination and showcase what is carried out well)	Lobby with unified voice to policy makers Help change culture—need to put horse first Support/inform social license to operate (by upskilling individuals/showcasing best practice) Showcase welfare Define what is best practice Disseminate to the Federation Equestre Internationale (FEI), national federations (NF) and member bodies (MB), and translate to inform officials and membership Share (positively) with media
Generate evidence (inform best practice and generate debate)	Inform research/next steps by identifying areas where further work/more information is needed Support and inform education initiatives NFs/MBs Put evidence and experience into practice Do not use [results] negatively to make money out of this topic Increased publication—different formats that translate evidence to all stakeholders Initiate research to generate evidence to inform practice

**Table 11 animals-13-03404-t011:** Participants’ perspective on key opportunities to improve sporthorse health and welfare.

Credibility	Strong will of core stakeholders: riders, grooms, and vets (provides a positive base to build on) Managing language—consider perception of language across equestrian communities and for the public (e.g., different groups of people will use different terminology and how they/others interpret this can affect individual and broader public understanding) Ensure horse (welfare) does come first
Education	Stakeholders: for riders, grooms, trainers, owners, officials to inform practice Public: [wider education] on practice and breadth of how horses are managed Recognition that practical use of horses (e.g., in competition) can result in transient reduction in some essential areas (such as turnout, social contact) Need to emphasise that horses enjoy athletic activities and that horses can indicate what they do not like, giving an opportunity (for humans) to change practice Focus on all levels/aspects of industry and competition Retirement management/practice Education (compulsory) [for equestrian stakeholders at all levels] on welfare, behaviour, and communication Stable management Training to prepare for work and prevent injury (will then improve welfare) Assessment of horse fitness (physical/psychological) for work Vets (as key information/dissemination source) Individual responsibility supported by education
Regulation	Provide credibility Focus on enforcement of rules There are rules and there are consequences if these are breached Be prepared to modify competitions (all aspects) Government/national governing body guidelines for husbandry
Showcase best practice	Elite level/elite competition Demonstrate level of care/consideration of horse Reward good practice

## Data Availability

Due to the sensitive nature of these data, they are not available for open access.

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
