# Peer review of "A Delphi Study to Determine International and National Equestrian Expert Opinions on Domains and Sub-Domains Essential to Managing Sporthorse Health and Welfare in the Olympic Disciplines"

_animals, 2023, doi:10.3390/ani13213404_

Round 1
Reviewer 1 Report
Comments and Suggestions for Authors
Please find comments attached

Author Response
We would like to thank all reviewers for their time in reviewing our work and for their feedback which has helped us to increase the clarity of communication and quality of our work. It is much appreciated.
Reviewer 1
Thank you for your positive comments on our work and helpful suggestions, which we feel have improved the quality of our submission.
Introduction
L75-77 This sentence feels unfinished.
Our apologies, we have amended and linked to the next sentence to increase clarity
L81 amend to [4,8] – completed
L109 ‘In an area where emotions run high, such as the evaluation of animal welfare’: whose
perspective are you referring to – the public, welfare scientists, people involved in horse sport?
Amended to public
L114 Good point, do you have a reference for this?
Citation added to support
L171 replace ‘that whilst’ with ‘but whilst’- amended
Materials and Methods
198 It would be helpful to mention (as in Fig 1) that not all 117 participants had input into all of the
rounds of the Delphi – maybe you could explain why you think that was. Also, it would be useful
here to direct readers to Fig. 3 the breakdown of the spread of participants across the areas that you
have specified in lines 191-195.
Included sentence to outline this and linked reader to Fig 1 and 3 for further details
223-224 It would be helpful to state what these 12 suggested areas were here.
Thank you for your feedback, these have been included
Results
384-386 Interesting! Do you think that these differences in priorities were due to differences in
training/experience/tradition/habits within different roles or geographical areas? Perhaps explore
this a little further in section 4.1 of your discussion.
We agree it is really interesting and there could be many reasons underpinning this; we have expanded discussion in 4.1 to reflect this.
Figure 3 Do you think that the lower number of welfare professionals vs high numbers of those
employed at a regulatory body would have affected the priorities coming through in your results?
We believe this is likely the case, however we wanted to place the emphasis and majority input from participants working in roles at elite / international level sport – these participants should have received training / updates re health and welfare priorities and many roles e.g., vets one could expect to have a substantial welfare focus as well. Analysis of differences between roles did highlight some variation in ratings to welfare staff but not as many as maybe expected.
Table 7 I think the term ‘sustainability footprint’ could do with additional explanation as it could be
interpreted in a few different ways.
We are cautious to add explanations to sub domains as these were not included in the Delphi stages for these and we could unintentionally misinterpret how participants viewed this, therefore respectfully we have elected not to add an definition here.
Do you think that subdomains being repeated throughout different domains affected participant
rating of those areas (For example are participants likely to rate the subdomain consistently
regardless of which domain it is attached to or potentially feel that if the subdomain has already
been rated highly in another domain they would be likely to rate it lower when they see it
repeated?)
This is an interesting perspective; all stages of the Delphi included a question for feedback on the context / format and we didn’t get any feedback indicating this, but it could well be the case. We have added a discussion point to represent this in the limitations section.
L624 This is an interesting finding – ‘there is poor awareness across the equestrian industry on
equine welfare’ yet many of the subdomains were not rated as essential because participants felt
that they were already managed well. What would be your opinion on this juxtaposition?
A very interesting outcome of the study! We feel that there is a clear dichotomy in that stakeholders feel welfare is poorly understood, often as it is a complex multifactorial concept, but also feel strong regulation / guidance exists at top levels of sport to manage horse welfare. Anecdotally health and welfare quite often seem to be interchangeable, and welfare is perhaps measured as provision of factors essential to survive rather than embracing 5 domains model, wellbeing and providing horses with a good life - additional text added to optimising performance and welfare section in discussion to address this point.
Table 10 The formatting on this table needs attention. Also, further explanation is required to
understand what is meant by the point ‘Public on practice and breadth of how horses are managed’.
Similarly, for ‘Education (compulsory) on welfare, behaviour and communication’ – education for
who?
Our apologies, not quite sure what happened with the formatting here when converted to a PDF – rectified (hopefully). Additional context to support themes presented included in [xx].
L640 remove ‘well’
Removed
Discussion
677-679 Good point!
Thank you
L725 These additional statistics should also be reported in the results section.
We have included this information.
L831 I believe creating opportunities for horses to experience positive welfare, as stated here, is one
area that is more frequently overlooked.
Thank you, we agree
L852 This exemplifies the trade-off between the utility of an animal and its welfare – issues relating
to improving the physical welfare of a sporthorse may be beneficial to both the horse and the
associated humans (in terms of performance) and this link is clearer than say the potential benefits
resulting from increased mental wellbeing. Do you think there should be more research into how
improving mental wellbeing may also impact sporthorse performance as this may encourage more
stakeholders to pay attention to this aspect which is currently and has historically been very much
overlooked?
We agree, measuring the positive impact of improve mental wellbeing is more difficult to measure and articulate potentially while physical improvements can be more clearly aligned to performance benefits – we feel this is important and have added a discussion point into text to address this.
L892 Do you think this presents difficulty surrounding the best course of action going forwards to
bridge these gaps?
Yes, the disconnect in how experts value welfare assessment across different contexts pose problems as there is not one simple solution and different perspectives and actions will need to be implemented to achieve success / upskill different audiences. A comment related to this has been added to the manuscript.
L930 This idea of variability over time in a welfare continuum is mentioned by Mellor (2016)
Updating Animal Welfare Thinking: Moving beyond the “Five Freedoms” towards “A Life Worth
Living”.
Thank you for highlighting this, we have included.
965-967 This phrasing feels repetitive, it could be removed.
We have considered your feedback here but would like to retain the text as we have had previous feedback on drafts that reiterating key results / conclusions is beneficial due to the large volume of material presented.
Reviewer 2 Report
Comments and Suggestions for Authors
First, I would like to offer my apologies for the delay. Second, I would like to congratulate you all on this study which is an important one at this time. I enjoyed reading the paper.
Overall, there are very few changes need to be made. I do however want the post-hoc analyses on country and role to be presented in the methods and the results and not appear first in the discussion.
Simple summary: fine.
Abstract: The sentence that starts on line 54 ‘Participants felt…’ is not 100% clear. Are they wanting increased policy development, or just new policy development, same for regulation. I think the three ‘ands’ in this sentence make it unclear as to what participants felt. Perhaps two sentences would be better.
Introduction:
Line 73: starting ‘Recently societal….’ This sentence does not entirely make sense. Surely it should be: ‘Until recently, societal concern was focused on the welfare of racehorses…’ with the following sentence changed to ‘Concern has now shifted to…’
Line 80: Horse welfare does not change in relation to what we think -but what the horse experiences. As such, I would suggest this sentence is changed to: ‘How horse welfare is valued has been primarily determined by…’
Lines 155-157: a picky point, but why are some disciplines capitalised, and others not. This just jars with me as a reader. I suggest choosing one format -and repeating for all.
Methods:
Well described!
Line 286-289: content analysis : what methodology was used here to undertake content analysis? The references used are not methodology references.
Page 8:
Clear diagram -formatting has gone a bit weird in the text box on the left. Is it possible to have the diagram and it’s legend all on the one page? Might be one for the editors.
Results:
Each section is a little repetitive, but this is probably unavoidable and most importantly, the results are pretty clear.
Line 377-378: the countries that did not participate were those that have only participated in the Olympics once -Ireland stands out as not participating and having been involved in the Olympics multiple times as far as I am aware. Should this be mentioned? Again -it raises questions for the reader. Think about adding to the discussion but not essential.
Table 1: heading -just call them core domains rather than core areas/domains. You have defined domains nicely above and used it in the text ,so don’t go back on it for the table!
Line 407: table 1 subscript. I would like a list of those rejected that were subsequently integrated into the final domains please rather than just writing ‘some’. I appreciate that you can work this out if you go through the other tables -but for ease of the reader it would be good to list them here.
Comment: I found the content analysis themes coming out after the domains regarding Stable management and welfare assessment were not deemed essential utterly fascinating!
Table 10: formatting. As these are bullet pointed, could they be left justified rather than centre as it is hard to read.
Could there be a way of demonstrating how many of the participants mentioned each view shown in tables 9 and 10? Its seems a shame not to continue the semi-quantitative study through to the end.
Although I understand the method is used to find consensus, it would be interesting to know where there were differences. Did you look at whether or not for example your veterinarians were different to your equine researchers (this has certainly been a factor in other animal welfare Delphis that I am aware of)? If not, could this be something to add to the discussion?
Discussion:
Generally, very well written.
OK, I found the post-hoc analysis on lines 725 etc. I fundamentally disagree with this being the first time such analyses are mentioned. There must be a section -it can be brief, in the methods for the KWs and freq analysis it could be just entitled ‘post-hoc analyses’. These results should be presented in the results section. The alternative could be to produce a Results and Discussion section in one if the journal allows.
Line 738 ‘tended’ please give full stats information.
Line 742: please give full stats information, not just the P value.
I would like a brief discussion of the comments from the thematic analysis, especially that on line 418 (mental and behavioural needs can be difficult to accommodate). I wondered if this could fit into the paragraph starting Line 846, or that starting 902. Considering most animal welfare scientists agree that welfare is explicitly what the horse experiences -and therefore their mental and behaviour needs are of paramount importance, the idea that they cannot be accommodated is somewhat problematic.
Conclusions: well produced.
Author Response
We would like to thank all reviewers for their time in reviewing our work and for their feedback which has helped us to increase the clarity of communication and quality of our work. It is much appreciated.
First, I would like to offer my apologies for the delay. Second, I would like to congratulate you all on this study which is an important one at this time. I enjoyed reading the paper.
Overall, there are very few changes need to be made. I do however want the post-hoc analyses on country and role to be presented in the methods and the results and not appear first in the discussion.
Thank you for your kind words, we are pleased you feel our work has worth. We have included an additional section on the tests of difference in the results section – we did debate on including this at first but elected not to due to the volume of results already presented but with hindsight agree this change would be beneficial.
Simple summary: fine.
Abstract: The sentence that starts on line 54 ‘Participants felt…’ is not 100% clear. Are they wanting increased policy development, or just new policy development, same for regulation. I think the three ‘ands’ in this sentence make it unclear as to what participants felt. Perhaps two sentences would be better.
Thank you for your feedback, we have amended to hopefully improve clarity
Introduction:
Line 73: starting ‘Recently societal….’ This sentence does not entirely make sense. Surely it should be: ‘Until recently, societal concern was focused on the welfare of racehorses…’ with the following sentence changed to ‘Concern has now shifted to…’
Thank you for suggestion which we have adopted.
Line 80: Horse welfare does not change in relation to what we think -but what the horse experiences. As such, I would suggest this sentence is changed to: ‘How horse welfare is valued has been primarily determined by…’
We appreciate your point and feel it is both how we think which informs our actions and therefore impacts what the horse experiences and affects their welfare; however, we agree this was not clear in the original text, and feel your suggestion increases the clarity here, thank you.
Lines 155-157: a picky point, but why are some disciplines capitalised, and others not. This just jars with me as a reader. I suggest choosing one format -and repeating for all.
Not picky at all, we agree and apologise for the oversight on our behalf – amended.
Methods:
Well described! Thank you
Line 286-289: content analysis: what methodology was used here to undertake content analysis? The references used are not methodology references.
We utilised conventional, inductive content analysis – we have clarified this within the text. We also released reference #44 was repeated and incorrect and have included the original we wanted to link to – apologies for this oversight.
Page 8:
Clear diagram -formatting has gone a bit weird in the text box on the left. Is it possible to have the diagram and it’s legend all on the one page? Might be one for the editors.
Apologies – something went amiss with the formatting here. We will refer to the editors re citing of figures / tables and legends on the same page – we agree it would be beneficial.
Results:
Each section is a little repetitive, but this is probably unavoidable and most importantly, the results are pretty clear.
We fully appreciate your point. Across the project we have presented the work to participating stakeholders and have debated widely across the group and research team what is the best approach to presentation. The general consensus has been to apply this consistent approach.
Line 377-378: the countries that did not participate were those that have only participated in the Olympics once -Ireland stands out as not participating and having been involved in the Olympics multiple times as far as I am aware. Should this be mentioned? Again -it raises questions for the reader. Think about adding to the discussion but not essential.
We tried to engage Irish stakeholders on numerous occasions unfortunately with no success, but they are a notable top equestrian nation which are missing. We have elected to not name and shame so to speak as we feel this may be counterproductive to nations engaging with the work in the future.
Table 1: heading -just call them core domains rather than core areas/domains. You have defined domains nicely above and used it in the text ,so don’t go back on it for the table!
Thank you – another area with lots of debate around presentation! Happy to change.
Line 407: table 1 subscript. I would like a list of those rejected that were subsequently integrated into the final domains please rather than just writing ‘some’. I appreciate that you can work this out if you go through the other tables -but for ease of the reader it would be good to list them here.
We have highlighted respective areas in bold for clarity.
Comment: I found the content analysis themes coming out after the domains regarding Stable management and welfare assessment were not deemed essential utterly fascinating!
We agree, quite insightful and hopefully useful for future research in this field to consider.
Table 10: formatting. As these are bullet pointed, could they be left justified rather than centre as it is hard to read.
Apologies this is another area where the original formatting went astray – points have been justified to the left.
Could there be a way of demonstrating how many of the participants mentioned each view shown in tables 9 and 10? Its seems a shame not to continue the semi-quantitative study through to the end.
While this is an approach we could have taken, we elected to engage in a conventional content analysis approach applying an inductive methodology and feel introducing frequency tallies would detract from this.
Although I understand the method is used to find consensus, it would be interesting to know where there were differences. Did you look at whether or not for example your veterinarians were different to your equine researchers (this has certainly been a factor in other animal welfare Delphis that I am aware of)? If not, could this be something to add to the discussion?
We did undertake this analysis, presented later in discussion, as we didn’t want the focus to become a them vs us debate, based on reviewer feedback we have now summarized this in the results.
Discussion:
Generally, very well written.
Thank you for your kind feedback.
OK, I found the post-hoc analysis on lines 725 etc. I fundamentally disagree with this being the first time such analyses are mentioned. There must be a section -it can be brief, in the methods for the KWs and freq analysis it could be just entitled ‘post-hoc analyses’. These results should be presented in the results section. The alternative could be to produce a Results and Discussion section in one if the journal allows.
On reflection, we agree (see comment above) and have included sections in both methods and results to initially outline this element of the study.
Line 738 ‘tended’ please give full stats information.
This reflects a trend in the data not a significant difference, refer to previous sentence, which is why no stats information is provided.
Line 742: please give full stats information, not just the P value.
With the revision of the paper, this is now included in the results.
I would like a brief discussion of the comments from the thematic analysis, especially that on line 418 (mental and behavioural needs can be difficult to accommodate). I wondered if this could fit into the paragraph starting Line 846, or that starting 902. Considering most animal welfare scientists agree that welfare is explicitly what the horse experiences -and therefore their mental and behaviour needs are of paramount importance, the idea that they cannot be accommodated is somewhat problematic.
We agree and have included additional discussion as suggested.
Conclusions: well produced.
Thank you
Reviewer 3 Report
Comments and Suggestions for Authors
This is a well-written and important research study looking to determine factors that are critical to long term sustainability of equestrian sports. The methodology is soundly supported by accompanying references. Results are not surprising but can provide traction on a path to inclusivity and improved health and welfare of sport horses. My comments are only minor, but in general the discussion is quite long. There are very important points within the discussion but it might lend to better readability if the discussion could be shortened.
L156 – note that reining is also an FEI-level sport competed at the world championships
L161 – what exactly is meant by “Showing” as a discipline? This could be Western Showmanship, equitation, breed shows, so would be helpful to provide clarity here
L402 – are you referring to information presented in the supplementary file here? If so then perhaps make reference to that. Otherwise the reader wonders what the sub-domains are that you mention. Same comment for other areas of the manuscript that discuss the sub-domains.
L414-432 – this may fit better in the discussion although it flows well here to keep everything clear and organized
L578 – I wasn’t clear (or maybe I missed it) how the supplementary file relates to tables 2-8. Are the sub-domains in the tables those that emerged from the successive Delphi rounds? Are the sub-domains in the supplementary file the original suggestions from the first Delphi round?
L655 – I think it would be interesting to analyze if there was any consistency in opinions among the groups of participants – i.e. vets, NF, allied professionals, rider/trainer, researcher. Could you do a quick statistical test to look at that? There could emerge some elucidating results that may better explain the variation – e.g. maybe the variation occurred because of one or two groups. Maybe researchers and vets were more aware of methods to assess various aspects of sport horse health and welfare.
Oh, I see you have done this on L725 and on (good on you!). However the information on the Kruskal-Wallis analyses should appear in the methods and results. P-values should not appear in the discussion.
Comments on the Quality of English LanguageMinor grammatical review required. Below are some things I noted
L75-76 – this is a sentence fragment. Maybe combine with the previous sentence?
L81 – should be a comma between refs 4,8
L159 – federations does not need an apostrophe
L442 – only half of the word “essential” is in italics
L640 – I think this sentence means to read “They also felt that there was a need…” (delete the word “well”)
Author Response
We would like to thank all reviewers for their time in reviewing our work and for their feedback which has helped us to increase the clarity of communication and quality of our work. It is much appreciated.
This is a well-written and important research study looking to determine factors that are critical to long term sustainability of equestrian sports. The methodology is soundly supported by accompanying references. Results are not surprising but can provide traction on a path to inclusivity and improved health and welfare of sport horses. My comments are only minor, but in general the discussion is quite long. There are very important points within the discussion but it might lend to better readability if the discussion could be shortened.
Thank you for your positive feedback, we hope the work will help inform future work in this area, across federations and research. We take on board that the discussion is a little long, however we have been encouraged to lengthen and expand in parts from the other reviewers feedback.
L156 – note that reining is also an FEI-level sport competed at the world championships
Thank you for flagging this omission – which is now included.
L161 – what exactly is meant by “Showing” as a discipline? This could be Western Showmanship, equitation, breed shows, so would be helpful to provide clarity here
A footnote has been added to outline what showing can encompass.
L402 – are you referring to information presented in the supplementary file here? If so then perhaps make reference to that. Otherwise the reader wonders what the sub-domains are that you mention. Same comment for other areas of the manuscript that discuss the sub-domains.
The sub-domains referred to link to the areas participants identified as underpinning the core domain areas, not the supplementary information. We have added a comment to clarify.
L414-432 – this may fit better in the discussion although it flows well here to keep everything clear and organized
We have elected to retain here to help support clarity as previous feedback has been it helps the reader to follow the threads of the results.
L578 – I wasn’t clear (or maybe I missed it) how the supplementary file relates to tables 2-8. Are the sub-domains in the tables those that emerged from the successive Delphi rounds? Are the sub-domains in the supplementary file the original suggestions from the first Delphi round?
The sub-domains are not related to the supplementary file. Participants agreed and proposed the sub domains that relate to the core domains in the main manuscript in Stage 2 of the Delphi (line 302-304).
L655 – I think it would be interesting to analyze if there was any consistency in opinions among the groups of participants – i.e. vets, NF, allied professionals, rider/trainer, researcher. Could you do a quick statistical test to look at that? There could emerge some elucidating results that may better explain the variation – e.g. maybe the variation occurred because of one or two groups. Maybe researchers and vets were more aware of methods to assess various aspects of sport horse health and welfare.
Oh, I see you have done this on L725 and on (good on you!). However the information on the Kruskal-Wallis analyses should appear in the methods and results. P-values should not appear in the discussion.
Our apologies, we were attempting to reduce the methods / results but with hindsight have reflected this is not the clearest presentation and have added text to these sections.
Comments on the Quality of English Language
Minor grammatical review required. Below are some things I noted
L75-76 – this is a sentence fragment. Maybe combine with the previous sentence?
This has been amended – apologies for the oversight.
L81 – should be a comma between refs 4,8
Amended
L159 – federations does not need an apostrophe
Removed
L442 – only half of the word “essential” is in italics
Apologies – missed this one, amended!
L640 – I think this sentence means to read “They also felt that there was a need…” (delete the word “well”)
Removed
Reviewer 4 Report
Comments and Suggestions for Authors
there are some very minor comments/edits/questions on the attached document
generally speaking, I think the authors might have benefitted their audience by keeping many of the sentences shorter & easier to understand (but my concern is not great enough to send it back & ask for a full revision - this is based largely on the fact that the material is time-sensitive <my opinion>

Comments on the Quality of English Languageminor items - please see comments in text box above, also comment bubbles on the attached PDF
Author Response
Thank you for your feedback and for providing additional suggestions on the manuscript; these have been extremely helpful in improving our work.
Response to comments on annotated PDF.
Line 80: comma placed between citation numbers
Line 103: convince amended to ‘prove to’
Line 211: we selected the three Olympic disciplines as these are the disciplines which have the most sport horses participating in them
Table 1: stable and environment management is hard to understand
We are cautious to add definitions to domains in the tables, these are therefore respectfully we have elected not to add a definition here. Participant agreed definitions are provided at the outset of the presentation of each domain after this point which will hopefully help the reader to understand what the participants viewed this as representing.
Table 9: ‘Do not use negatively to make money out of this topic’ confusing – we have included [results] after use to aid clarity
Table 10: rephrased for clarity
Table 10: suggest not every horse enjoys every athletic activity – we would agree however the table reflects the themes derived from participant input
Table 10: showcasing practice could be at all levels of competition - we agree, however the table reflects the themes derived from participants who highlighted the elite level of the sport
Line 670: well has been removed
Line 695: On further questioning, the majority of participants felt that these areas ‘were not essential’ as they were already managed well – important difference worth expansion in discussion
Thank you for highlighting the importance of this area, we have expanded discussion as suggested to consider further.